# Bimetallic AgFe Systems on Mordenite: Effect of Cation Deposition Order in the NO Reduction with C$_3$H$_6$/CO

**Perla Sánchez-López** [1], **Yulia Kotolevich** [1,*], **Serguei Miridonov** [2], **Fernando Chávez-Rivas** [3], **Sergio Fuentes** [1] **and Vitalii Petranovskii** [1]

[1] Centro de Nanociencias y Nanotecnología, Universidad Nacional Autónoma de México, Ensenada 22860, Mexico; perlasanchezlopez23@gmail.com (P.S.-L.); fuentes@cnyn.unam.mx (S.F.); vitalii@cnyn.unam.mx (V.P.)

[2] Centro de Investigación Científica y de Educación Superior de Ensenada, Ensenada 22800, Mexico; mirsev@cicese.mx

[3] Escuela Superior de Física y Matemáticas, Instituto Politécnico Nacional, Ciudad de México 07738, Mexico; fchavez@esfm.ipn.mx

* Correspondence: julia.kotolevich@gmail.com; Tel.: +52-646-175-0650

**Abstract:** Mono- and bimetallic systems of Ag, Fe, and Ag–Fe exchanged in sodium mordenite zeolite were studied in the reaction of NO reduction. The transition metal cations Ag and Fe were introduced by ion exchange method both at room temperature and 60 °C; modifying the order of component deposition in bimetallic systems. These materials were characterized by Inductively Coupled Plasma-Optical Emission Spectroscopy (ICP-OES), ultraviolet-visible spectroscopy (UV-Vis), X-Ray photoelectron Spectroscopy (XPS) and High-resolution transmission electron microscopy (HR-TEM). The XPS and UV–Vis spectra of bimetallic samples revealed that under certain preparation conditions Ag$^+$ is reduced with the participation of the Fe$^{2+}$/Fe$^{3+}$ ions transition and is present in the form of a Ag reduced state in different proportions of Ag$_m$ clusters and Ag$^0$ NPs, influenced by the cation deposition order. The catalytic results in the NO reduction reaction using C$_3$H$_6$/CO under an oxidizing atmosphere show also that the order of exchange of Ag and Fe cations in mordenite has a strong effect on catalytic active sites for the reduction of NO.

**Keywords:** mordenite; AgFe bimetallic; ion exchange; NO reduction

## 1. Introduction

In recent decades, there has been considerable interest from regulatory organizations and the scientific community in solving environmental issues associated with NO$_x$ emissions. Atmospheric NO$_x$ are known to be the primary precursors of secondary pollutants, such as nitric acid, ozone, and peroxyacyl nitrates, which are responsible for effects as photochemical smog, tropospheric ozone, and acid rain, among others [1]. As a result, various technologies and processes for removing NO$_x$ emissions from exhaust gases have been developed [2–5].

The selective catalytic reduction of NO$_x$ (SCR of NO$_x$) with ammonia or light hydrocarbons as reducing agents is one of the promising ways to remove nitrogen oxides from mobile exhaust sources [6–9]. As catalysts for NO$_x$ reduction, are used transition metal cations and multi-metallic mixtures (Fe$^{3+}$, Co$^{2+}$, Ni$^{2+}$, Cu$^{2+}$, Zn$^{2+}$, Ag$^+$, and others), supported on various carriers, including zeolites [10–15]. The unique physicochemical properties of the zeolites, such as their controlled acidity, adsorption capacity, ion exchange properties, and thermal stability, as well as uniform channels and cavities crystallographically ordered in size and position determine their effectiveness in catalytic

processes [16,17]. Thus, zeolites being a very suitable substrate to form clusters and small nanoparticles either outside or inside the cavities, can be a great option as supports for active metals in SCR of $NO_x$ [18–20]. Chen et al. [21] analyzed the reduction of $NO_x$ over Fe/zeolite catalysts with *iso*-butane and propane. They showed that the catalytic activity decreases as follows: Iron on beta zeolite (Fe/BEA) > iron on ZSM-5 zeolite (Fe/MFI) > iron on ferrierite zeolite (Fe/FER) > iron on mordenite zeolite (Fe/MOR) ≈ iron on Y-zeolite (Fe/FAU). Such order of activity was caused by the structural characteristics of the zeolites, i.e., the dimensionality and size of pores. Zeolites with open cavities smaller than $4.2 \times 5.4$ Å and $3.5 \times 4.8$ Å were able to oxidize NO to $NO_2$ formed from $NO_x$ complexes, but they blocked the entrance of reductive molecules.

Some studies [22,23] have shown that zeolite-based multi-metallic materials exhibit a synergistic effect in comparison with monometallic materials. The interaction of Ag and Fe supported on different zeolite carriers has been the subject of many modern studies because it affects the catalytic [24], ferromagnetic [25], electronic [26], luminescence [27], and other properties. Being relatively inexpensive materials [28], Ag-based bimetallic materials with transition metal additives have acquired a great interest, showing excellent catalytic activities for reduction of $NO_x$ [29,30].

A recent study of bimetallic catalysts [31] shows that the order of metal introduction can be a factor affecting their catalytic properties. Thus, the order of component deposition in binary materials has been proposed as a parameter to improve the catalytic activity and modify the properties [32–35]. Moreover, particular attention has been paid to the study of certain parameters in catalysts for the process of catalytic reduction of $NO_x$, such as the metal exchange degree [14,36], the framework type [37], the Si/Al ratio [38–40], the cationic chemical composition [41,42] and the reaction conditions.

The aim of this work was to study the catalytic activity of bimetallic AgFe/MOR catalysts in the $NO_x$ reduction, and to analyze the influence of the order of cation deposition.

## 2. Results and Discussion

The monometallic Ag- and Fe-containing samples were prepared by ion-exchange with double excess of cations and labeled AgMOR or FeMOR. The bimetallic systems were prepared in three different ways varying the order of incorporation of $Ag^+$ and $Fe^{2+}$ cations as follows: (1) Single stage ion-exchange from a binary mixture solution when Ag and Fe-containing solutions were mixed in a volume ratio of 1:1 (labeled mAgFeMOR); (2) double-stage ion-exchange, Ag first, then Fe (labeled AgFeMOR); (3) double-stage ion-exchange, Fe first, then Ag (labeled FeAgMOR). The effect of temperature on the ion exchange at ambient temperature (Ta) and 60 °C was studied also and is indicated in the sample labels (for example, $AgMORT_a$ and $AgMORT_{60}$). Table 1 shows ion-exchange preparation parameters, which include the theoretical ion exchange capacity, determined by the Si/Al ratio.

**Table 1.** Ion-exchange solutions preparation parameters of the exchange while using 3 g of NaMOR for the sample preparation, and the theoretical limit of silver ($Ag^+$) and iron ($Fe^{2+}$) content in the samples.

| Sample | Ion Exchange, First Step | | | Ion Exchange, Second Step | | |
|---|---|---|---|---|---|---|
| | Solution, 0.1 N | Volume, mL | Calculated Ion Exchange Capacity, Atomic % | Solution, 0.1 N | Volume, mL | Calculated Ion Exchange Capacity, Atomic % |
| FeMOR | $Fe(ClO_4)_2$ | 104 | 3.2 | - | - | - |
| AgMOR | $AgNO_3$ | 104 | 6.4 | - | - | - |
| AgFeMOR | $AgNO_3$ | 52 | 6.4 | $Fe(ClO_4)_2$ | 52 | 3.2 |
| FeAgMOR | $Fe(ClO_4)_2$ | 52 | 3.2 | $AgNO_3$ | 52 | 6.4 |
| mAgFeMOR | $AgNO_3$ + $Fe(ClO_4)_2$ | 52 + 52 | (Ag + 2Fe) = 6.4 | - | - | - |

## 2.1. Physicochemical Properties

The elemental composition of samples and their charge balance are shown in Table 2. The Si/Al ratio of all mono- and bimetallic samples remains constant and is in good agreement with the value for NaMOR before the exchange treatment (Si/Al = 6.5 ± 0.2) indicating that dealumination does not occur. In the commercial NaMOR used as a support, the nominal cation of compensation of a negative charge in the zeolite matrix is $Na^+$. According to the zeolite structural formula, the ratio of $Na_2O/Al_2O_3$ or simpler form Na/Al (known in industry as the caustic module) for the equilibrium structure of the zeolite should be equal to 1 [43,44]. In our case, the caustic module in commercial NaMOR was 1.29. The 0.29 excess may be attributed to the fact that synthesis of zeolites occurs in alkaline solutions, with a high caustic module [45], and occlusion of the mother liquor in the resulting crystals could be observed [46–48]. The Table 2 also shows the total concentrations of cations in the samples, which were calculated under the assumption that Ag is in oxidation state (I) and Fe may be in two oxidation states (II and III); charge balance requires that the total number of positive and negative charges in a zeolite sample be the same. The sum of the contents of cations that compensate for the negative charge in the zeolite lattice can be called the equilibrium ion-exchange modulus (EIEM) and it should be equal to one. After the ion exchange with Ag and Fe this ratio decreased for both cases $Fe^{2+}$ and $Fe^{3+}$ (See charge balance in Table 2). In the first case ($Fe^{2+}$), the caustic module is close to 1.0 (from 1.14 to 0.90). In the second case ($Fe^{3+}$), the module is closer to the nominal module of the NaMOR (1.25 to 1.06). We assume that both $Fe^{2+}$ and $Fe^{3+}$ valence states coexist in bimetallic samples, and the ratio between them depends on the order of exchange of Fe and Ag cations. The decrease of the caustic module may occur due to additional washing of samples in line with the replacement of $Na^+$ cations by $Fe^{2+}$ and $Ag^+$ during the exchange, or by exchange of protons ($H^+$) since the exchange procedure is carried out in a weakly acid environment at pH = 4.5. It has to be noted that increase of the Brønsted acidity of the mordenite is typically observed during the exchange reaction [49]. Thus, the protonation of exchange sites may occur during samples preparation in the course of the exchange of $Ag^+$ and $Fe^{2+}$, causing a charge imbalance in the mordenite. At the same time, $Ag^+$ can be reduced due to the metal-support interactions. Despite the fact that the ion exchange of $Fe^{2+}$ takes place at a slightly acidic pH, a certain amount of $Fe^{2+}$ can be oxidized to the state of $Fe^{3+}$. Thus, it is necessary to take into account the oxidation state of metals ($Ag^+$, $Fe^{2+}$ and $Fe^{3+}$) in the charge balance.

**Table 2.** Elemental composition by ICP-OES of studied samples.

| Sample | Ion Exchange Temperature | Atomic % | | | | | | | Charge Balance | |
|---|---|---|---|---|---|---|---|---|---|---|
| | | Si | Al | Na | Ag | Fe | Si/Al | Na/Al | EIEM-$Fe^{2+}$ | EIEM-$Fe^{3+}$ |
| NaMOR | - | 48.9 | 7.5 | 9.7 | - | - | 6.5 | 1.29 | 1.29 | - |
| AgMOR | ambient | 42.8 | 6.8 | 2.4 | 4.9 | - | 6.3 | 0.35 | 1.07 | - |
| AgFeMOR | | 41.7 | 6.3 | 1.0 | 3.8 | 0.9 | 6.6 | 0.16 | 1.05 | 1.19 |
| FeAgMOR | | 39.9 | 6.1 | 1.1 | 4.4 | 0.6 | 6.5 | 0.18 | 1.10 | 1.20 |
| mAgFeMOR | | 41.1 | 6.3 | 1.7 | 3.9 | 0.6 | 6.5 | 0.27 | 1.08 | 1.17 |
| FeMOR | | 44.2 | 6.9 | 4.0 | - | 1.1 | 6.4 | 0.58 | 0.90 | 1.06 |
| AgMOR | 60 °C | 40.8 | 6.4 | 1.3 | 5.4 | - | 6.4 | 0.20 | 1.05 | - |
| AgFeMOR | | 40.1 | 6.3 | 1.0 | 3.5 | 0.9 | 6.4 | 0.16 | 1.00 | 1.14 |
| FeAgMOR | | 41.9 | 6.4 | 1.5 | 4.4 | 0.7 | 6.5 | 0.23 | 1.14 | 1.25 |
| mAgFeMOR | | 41.7 | 6.4 | 1.6 | 4.0 | 0.7 | 6.5 | 0.25 | 1.09 | 1.20 |
| FeMOR | | 46.7 | 7.3 | 4.1 | - | 1.3 | 6.4 | 0.56 | 0.92 | 1.09 |

The content of silver and iron is variable for all samples (see content of the elements in Table 2). Monometallic samples of silver at both exchange temperatures represented the highest concentration (4.9 atomic % at 25 °C and 5.4 atomic % at 60 °C) which is close to the calculated capacity for the exchange of sodium cations in mordenite (6.4 atomic % of ion exchange capacity for $Ag^+$ according to theoretical formula $Na_{6.4}Al_{6.4}Si_{41.6}O_{96}$, $Ag_{6.4}Al_{6.4}Si_{41.6}O_{96}$). The Fe-monometallic samples contain 1.1 atomic % at room temperature and 1.3 atomic % at 60 °C which are lowers than the theoretical

capacity of 3.2 atomic % for $Fe^{2+}$ ($Fe_{3.2}Al_{6.4}Si_{41.6}O_{96}$). The results for bimetallic systems with different order of cation incorporation show that the second incorporated metal increases its exchange fraction, e.g., 3.8 to 4.4 atomic % for silver in $AgFeMORT_a$ and $FeAgMORT_a$, respectively. This is because during the second exchange, the sites occupied by both sodium and the first exchanged cation are replaced by the second exchanged cation. The same effect on the order of exchange observed for Ag in bimetallic samples was also observed for Fe. As for temperature, it is clear that this parameter does not strongly affect the quantity of the metal being exchanged. In accordance with our previous results [50], increasing the temperature of ion exchange, we expected an increase in the content of Ag and Fe cations due to the acceleration of diffusion, which was actually observed for monometallic samples. However, in a bimetallic catalyst, competition for the exchange sites plays a critical role cause of the cation sizes are close to the diameter of the zeolite channel. Basing on ICP-OES results it can be concluded that for bimetallic samples the experimental silver loading is diminished by the presence of iron and the Ag cations are easily exchanged into the mordenite matrix, while Fe cations have a lower exchange affinity.

UV–Vis spectroscopy was applied to identify the silver and iron species. Figure 1a,b shows the spectra of mono- and bimetallic systems prepared at room temperature and 60 °C. The absorption spectra of Ag-monometallic samples show three main bands associated with: (1) $Ag^+$ ions, (2) silver clusters, and (3) Ag nanoparticles. The peak with a maximum at 208 nm, and the shoulder around 222 nm correspond to the $Ag^+$ electronic transitions $[Kr] 4d^{10} \rightarrow [Kr] 4d^9 5s^1$ of isolated silver ions located on the ion exchanged sites of the mordenite framework [51–53]. The other three weak bands observed, two centered at 280 and around 325 nm, and the last one initiating about 370 nm and stretched up to 700 nm are attributed to the small silver cationic clusters $Ag_m^{n+}$ (3 < m < 5), metal cluster $Ag_m$ (m $\leq$ 8) and silver metallic nanoparticles, respectively [54,55]. The broad band from 370 nm to 700 nm is interpreted as the formation of silver nanoparticles in the range of <2 nm. In the case of FeMOR samples prepared at $T_a$ and $T_{60}$, the spectra show mainly a band with a maximum centered at 270 nm, which is attributed to oxygen-to-iron charge transfer of isolated $Fe^{3+}$ ions in tetrahedral or higher coordination ($Fe^{3+}O_4$ and $Fe^{3+}O_{4+x}$ (x = 1, 2)) [56–60]. Additionally, two more contributions took place: 340 nm, which is related to octahedral Fe ions in oligomeric clusters of the $Fe_xO_y$ type inside the mordenite channels, and 470 nm, corresponding to the $Fe_2O_3$ particles located on the external surface of mordenite particles [56].

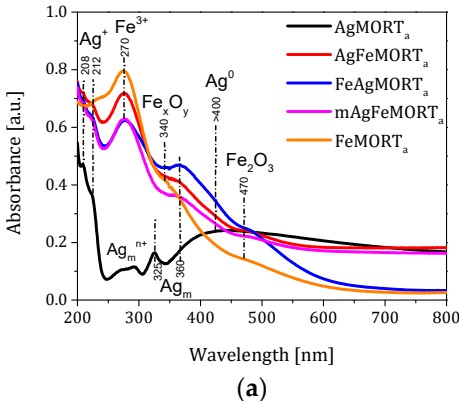 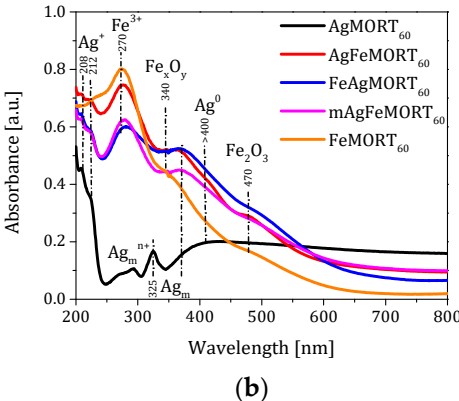

(a)　　(b)

**Figure 1.** UV–Vis spectra of mono- and bimetallic samples prepared at: (**a**) room temperature, and (**b**) 60 °C.

In bimetallic systems the presence of these species of Ag and Fe was noticeable; in particular, the contribution of $Fe^{3+} \leftarrow O$ charge transfer at 270 nm stands out significantly. Only for AgFeMOR, FeAgMOR and mAgFeMOR bimetallic systems (at both temperatures) a wide band was observed starting around nm and extended up to 450 nm, which was assigned to silver metallic nanoparticles. Such a band was not observed in the $FeMORT_a$ and $FeMORT_{60}$ spectra; so, we can infer that the

presence of $Fe^{3+}$ promotes the formation of reduced silver species. In addition, the intensity of the band of electronic transitions of $Ag^+$ ions about 208 nm decreases in the presence of iron, confirming the reduction of $Ag^+$ ions.

Thus, we conclude that in the bimetallic samples the corresponding Ag nanoparticles of size lower than 2 nm are formed by a redox process between $Ag^+$ and $Fe^{2+}$, which is manifested by the higher intensity of the band at 270 nm. Therefore, when $Ag^+$ is the first one exchanged it is reduced by the second exchanged cation, that is iron $Fe^{2+}$, forming silver metallic nanoparticles of smaller dimensions as compared to the auto-reduction process of $Ag^+$ in AgMOR.

XPS measurements were performed to study the oxidation state of silver and iron on the surface of the samples. All samples showed the characteristic Ag 3d and Fe 2p bands. In the case of Ag, the $3d_{5/2}$ and $3d_{3/2}$ photoelectron peaks presented very good resolution; in the case of Fe, the low metal content gives weak signal of $2p_{3/2}$ and $2p_{1/2}$ peaks. The binding energy (BE) measured from the mean position of deconvoluted bands of Ag $3d_{5/2}$ signal indicate the presence of different oxidation states on the surface (Table 3, Figures 2 and 3). The $3d_{5/2}$ peak position of the main component (80%) attributed to Ag-monometallic samples was $\geq$369.0 eV, which corresponds to $Ag^0$. This band showed a shift of 3d peaks to higher energies (0.3–0.9 eV) in bimetallic samples, which is related to small silver nanoparticles with a size lower than 2 nm [61,62]. The peak position of Fe $2p_{3/2}$ for Fe-monometallic samples was about 711.0 eV, this BE is related to $Fe^{3+}$ iron oxides ($Fe_2O_3$) [61]. In Fe-containing bimetallic systems, the band is shifted to higher BE in the range of 0.5–0.7 eV, revealing the presence of iron in the intervalent state ($Fe_3O_4$ from mixed $Fe^{2+}/Fe^{3+}$). In conclusion it has to be mentioned that the oxidation state of Ag and Fe in bimetallic samples strongly depends on the metal loading and the deposition order. For all samples the BE from the Ag spectra showed that ionic silver is reduced and is present mainly on the surface as Ag-NPs and iron appears like oxides in intermediate states in both cases.

**Table 3.** Mean peak position of the de-convoluted bands from Ag 3d XPS spectra.

| Sample | Ag 3d$_{5/2}$ Binding Energy [eV] | | | |
|---|---|---|---|---|
| | Ag$^0$ (%) 368.0–368.3 eV | Ag < 2 nm (%) $\geq$ 369.0 eV | Ag$^+$ (%) 366.2 eV | Ag-Support (%) 367.4–368.0 eV |
| AgMORT$_a$ | - | 369.0 (80) | | 367.0 (20) |
| AgFeMORT$_a$ | - | 369.7 (78) | - | 367.7 (22) |
| FeAgMORT$_a$ | 368.8 (80) | | | 366.8 (20) |
| mAgFeMORT$_a$ | - | 369.0 (80) | | 367.0 (20) |
| AgMORT$_{60}$ | - | 369.3 (82) | - | 367.5 (18) |
| AgFeMORT$_{60}$ | 368.8 (84) | | | 367.0 (16) |
| FeAgMORT$_{60}$ | - | 369.0 (82) | - | 367.8 (18) |
| mAgFeMORT$_{60}$ | - | 369.3 (74) | - | 367.3 (26) |

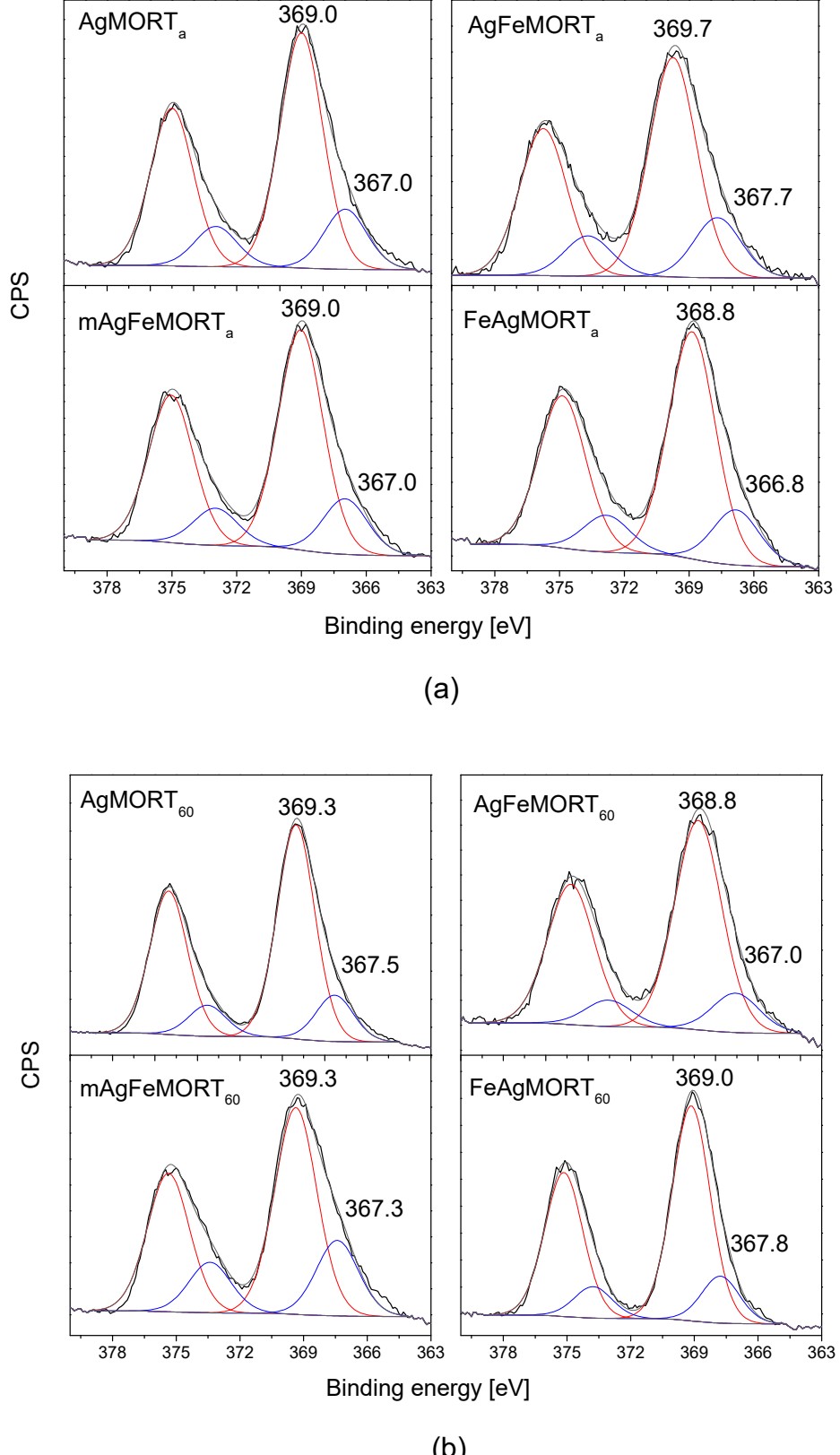

**Figure 2.** Ag $3d_{5/2}$ photoelectron spectra of mono- and bimetallic samples prepared at: (**a**) room temperature, and (**b**) 60 °C.

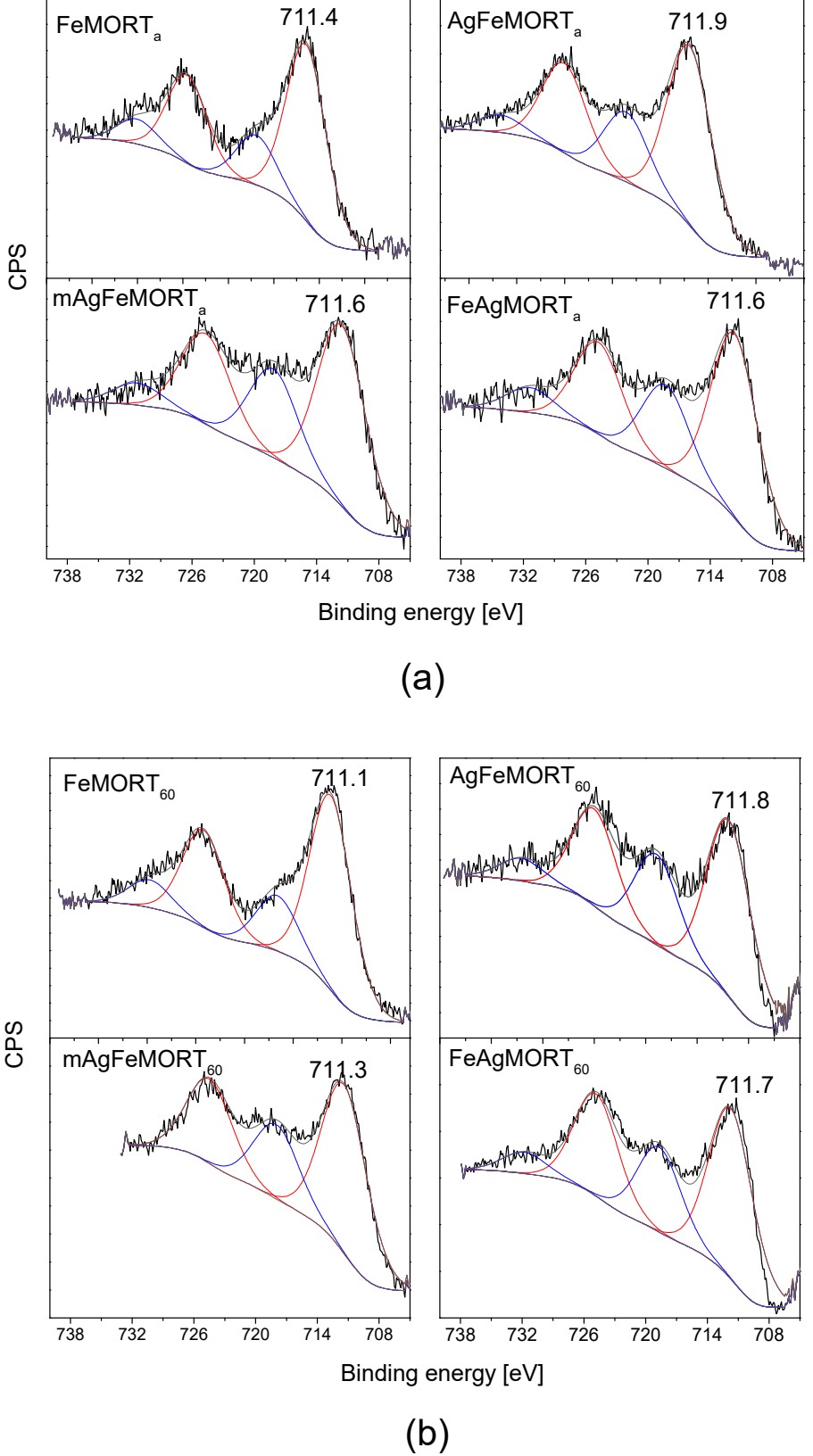

**Figure 3.** Fe 2p$_{3/2}$ photoelectron spectra of mono- and bimetallic samples prepared at: (**a**) room temperature, and (**b**) 60 °C.

Figure 4 shows representative micrographs of catalysts and particle size distributions. The presence of silver nanoparticles on the surface is clearly observed in micrographs of AgMORT$_{60}$, AgFeMORT$_{60}$ and mAgFeMORT$_{60}$ (Figure 4a,c,d, respectively). The average particle diameter of the catalysts is shown in Table 4. The average particle diameter in the Ag-containing samples is in the range of 3.0–6.5 nm. According to the particle size distribution in the histograms of bimetallic systems, a decrease in particle size is observed compared to Ag-monometallic samples (e.g., 4.4 and 6.2 nm in AgMOR change to 2.8 and 3.4 nm in FeAgMOR at both temperatures, respectively). Thus, the agglomeration of silver is significantly different in the presence of iron in bimetallic samples. In this work, no reduction treatments were undertaken; cationic silver is reduced by oxidizing Fe(II) to Fe(III). For AgFe and FeAg bimetallic systems the reduction in particle size was independent on the order of deposition of the cations, this is due to the fact that iron (II) species interact with silver (I) cations, causing the formation of smaller silver nanoparticles (Ag NPs). In the case of iron, the micrographs of the FeMORT$_{60}$ sample confirm that Fe cations are well dispersed in the zeolite framework without agglomeration.

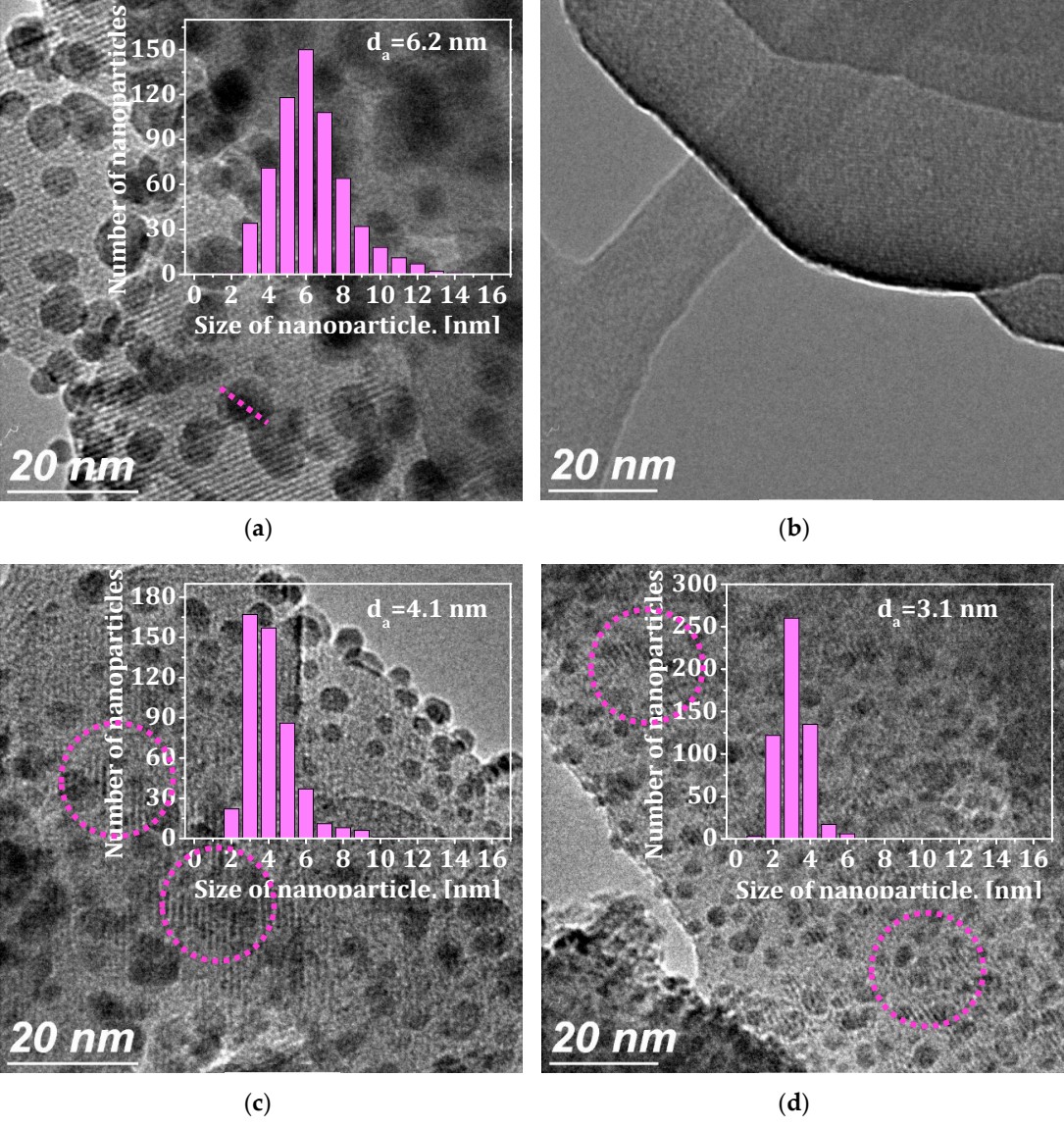

**Figure 4.** HRTEM-Micrographs and particle size distribution of the samples: (**a**) AgMORT$_{60}$, (**b**) FeMORT$_{60}$, (**c**) AgFeMORT$_{60}$, and (**d**) mAgFeMORT$_{60}$.

**Table 4.** Average diameters of silver particles of the studied mordenite-based samples.

| Catalyst | Temperature of the Ion Exchange | Average Particle Diameter, nm |
|----------|-------------------------------|-------------------------------|
| AgMOR | ambient | 4.4 |
| | 60 °C | 6.2 |
| AgFeMOR | ambient | 3.8 |
| | 60 °C | 4.1 |
| FeAgMOR | ambient | 2.8 |
| | 60 °C | 3.4 |
| mAgFeMOR | ambient | 6.8 |
| | 60 °C | 3.1 |

*2.2. Catalytic Conversion of NO with $C_3H_6$*

In general, the established reaction conditions for SCR of NO in presence of hydrocarbons under oxidizing conditions is performed in the range from 500–2000 ppm of NO, under oxidizing atmosphere (1–3% of $O_2$) and 400–2500 ppm of reducing agent ($C_3$-hydrocarbons; mainly $C_3H_8$ and $C_3H_6$). For such conditions maximum values of NO conversion about 60–90% and selectivity to nitrogen around 70% are obtained in the temperature range 350–450 °C [2–15,63].

In this work, the NO conversion was performed with $C_3H_6$/CO in oxidizing atmosphere (2.1% vol $O_2$) for all catalysts and the conversion profiles are shown in Figure 5. In this figure, the catalytic activity of AgFeMOR and FeAgMOR bimetallic catalysts is compared with those of Ag or Fe monometallic catalysts. Results show that the bimetallic catalysts (AgFeMOR, FeAgMOR, and mAgFeMOR), prepared at both $T_a$ and $T_{60}$ reach a maximum of NO conversion (about 70–90%) in the temperature range of 300–475 °C (Figure 5b,d), meanwhile, AgMOR and FeMOR monometallic catalysts (both $T_a$ and $T_{60}$) presented much less activity, about 25% at 500 °C and 50% at 300 °C, respectively (Figure 5a,c). Additionally, the initial non-exchanged NaMOR itself showed catalytic activity with maximum (12.4%) at 300 °C which is probably due to $Na^+$ cation that acts as a Lewis acid site, while the oxygen framework with partial negative charge acts as a Lewis base [64].

The AgMOR monometallic catalysts at both temperatures presented low-temperature activity (~125 °C), which is attributed to the presence of $Ag^+$ ions catalyzing a mild oxidation process, since the propene is not yet activated [36]. By increasing the temperature to 350–520 °C a further increase in conversion is observed which is attributed to silver in reduced state ($Ag^+ \rightarrow Ag_m^{n+} \rightarrow Ag_m \rightarrow Ag^0$ NPs) [65–68]. Such reduced species of silver were generated during the ionic exchange according to UV–Vis spectra (Figure 1a,b). In general, the Ag zeolite catalysts present low activity in NO conversion to $N_2$ (around 20–40%) [51,68]. The active sites proposed by Aspromonte et al. [52] for AgMOR catalysts in the SCR de $NO_x$ with toluene and butane, were different Ag species: ($Ag^+$ ion, $Ag_m^{n+}$ cluster and $Ag^0$ NPs), and the metal loading is an important parameter for the catalytic activity. They also reported that the NO reduction decreased for high levels of metal exchanged for AgZSM5 catalyst.

The FeMOR monometallic catalysts ($T_a$ and $T_{60}$) showed about 50% conversion of NO at 300 °C (see Figure 5a,c) which according to the UV–Vis spectra can be attributed to $Fe^{3+}$ mononuclear species stabilized in the mordenite framework, similarly to the case of Fe-catalyst for $NO_x$ reduction with $NH_3$ where the active sites are monomeric iron sites (Fe-O-OH at 300 °C) [69]. Other reports also have shown that Fe/zeolites catalysts exhibit high catalytic activity in NO and $N_2O$ reduction process [70–72].

Figure 5b shows the conversion profiles of $NO_x$ for bimetallic catalysts prepared at room temperature. The AgFeMOR$T_a$ and FeAgMOR$T_a$ presented two peaks of $NO_x$ conversion each, at 310 and 400 °C and 350 and 428 °C, respectively, meanwhile, the mAgFeMOR$T_a$ showed only a maximum at 375 °C. The three bimetallic catalysts prepared at 60 °C (Figure 5d), AgFeMOR$T_{60}$ (305 and 395 °C), FeAgMOR$T_{60}$ (370 and 467 °C) and mAgFeMOR$T_{60}$ (365 °C) presented similar behavior that those prepared at $T_a$.

In these bimetallic catalysts (prepared both at $T_a$ and $T_{60}$), the maxima for $NO_x$ conversion due to active sites of $Fe^{3+}$ species (around 300 °C) were shifted about 10–60 °C with respect to the monometallic FeMOR catalyst, revealing an interaction between silver and iron. Such interaction can be related to the reduction of Ag ions observed in UV–Vis spectra ($Ag^+ \rightarrow Ag_m{}^{n+} \rightarrow Ag_m \rightarrow Ag^0$ NPs). Thus, the $NO_x$ conversion due to reduced Ag species in bimetallic catalysts at both temperatures of preparation increased notoriously with respect to the AgMOR monometallic catalyst reaching more than 60% in the range of 350–500 °C.

The monometallic catalysts prepared at room temperature and 60 °C presented different loadings of exchanged metal, the amount of silver (4.9–5.4 atomic %) was about four times higher than the amount of iron (1.1–1.3 atomic %). Reports of Fe/zeolites catalysts showed that the level of Fe exchange is generally low, but the catalytic performance of such catalysts is very good (Gurgul et al. [58]). They reported about the effect of total Fe content, and the iron species that were presented in FeBEA samples; for systems with a low Fe content, the active specie was $Fe^{3+}$ in tetrahedral coordination; then our results are in very good agreement with these results of the samples containing iron, in which the main catalytic specie is $Fe^{3+}$.

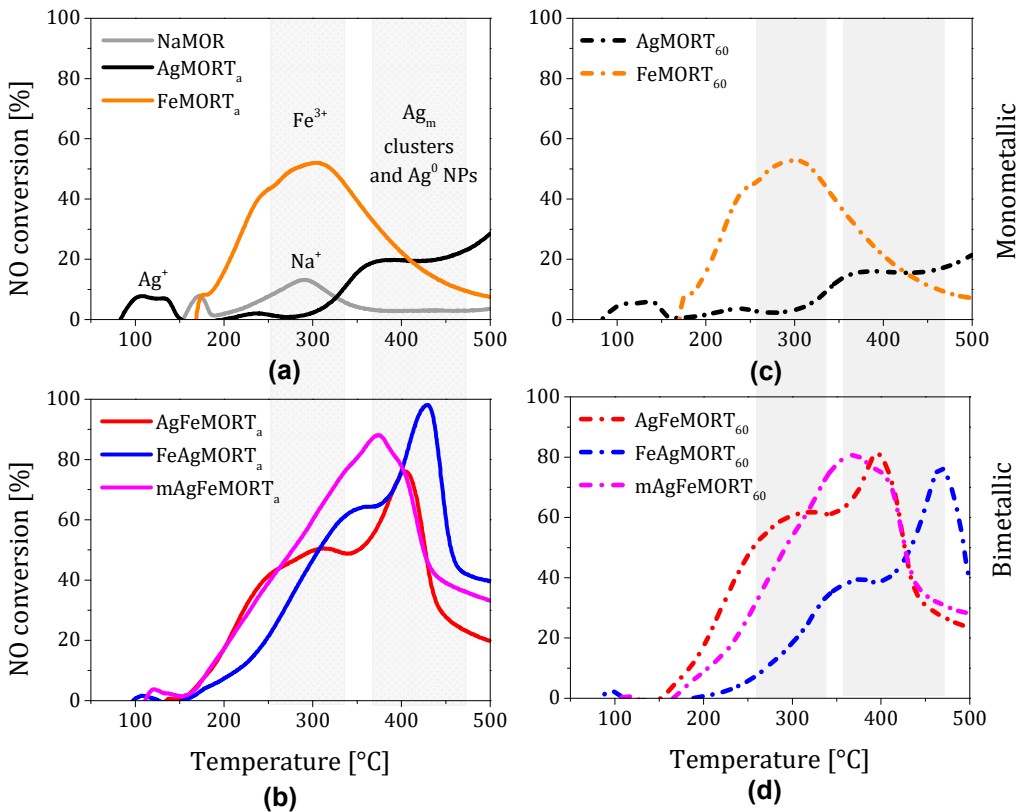

**Figure 5.** Catalytic performance of NaMOR, Ag or Fe monometallic, and Ag-Fe bimetallic catalysts prepared at room temperature (**a**,**b**) and 60 °C (**c**,**d**) for NO reduction as a function of reaction temperature. Reaction conditions: NO (850 ppm), $C_3H_6$ (350 ppm), $O_2$ (2.1 vol %), CO (0.4 vol %), and $N_2$ balance, flow rate 78 mL·min$^{-1}$ and GHSV (15,000 h$^{-1}$).

## *2.3. Effect of Cations Deposition Order on NO Conversion*

The catalytic behavior of bimetallic catalysts shows the existence of a strong influence of the order of deposition of cations on the NO conversion as a function of the reaction temperature. The Figure 5b,d shows the profiles of $NO_x$ conversion for the AgFeMOR, FeAgMOR, and mAgFeMOR bimetallic catalyst prepared at $T_a$ and $T_{60}$. The incorporation of iron into the AgMOR catalyst (AgFeMOR at both temperatures) stabilized the $Fe^{3+}$ sites in the mordenite framework at the same time as it promoted the formation of oxidative Ag species (Ag$_m$ cluster and the presence of $Ag^0$ nanoparticles). This is in

agreement with the UV–Vis and XPS results showing that the main species of Ag or Fe in bimetallic catalysts were $Ag^0$ NPs, $Ag_m$ metallic clusters, and $Fe^{3+}$, respectively. In the case of the incorporation of silver in the FeAgMOR system, the presence of the same active sites is also observed, but the maxima of their catalytic activity is shifted to higher temperatures.

The FeAgMORTa catalyst presents higher $NO_x$ conversion than the catalyst prepared a $T_{60}$; this difference could be caused by the presence of Ag nanoparticles with sizes higher than 3 nm as shown in Table 4. The mAgFeMOR system, prepared by ionic exchange competition has only one active site at an intermediate temperature range (380 °C) with respect to the maxima of NO conversion temperature for the Ag and Fe monometallic catalysts. Then, a synergistic effect is clearly observed in this mAgFeMOR catalyst (both at $T_a$ and $T_{60}$).

The promoter effect of a second transition metal has been reported previously, for instances, various metals as Cu, Pd, Fe, Ni, or Co. were added to Ag-catalysts, supported on $Al_2O_3$ and zeolites in order to improve their catalytic activity [10,30,73]. Other results show that the incorporation of iron as the second metal promotes the formation of silver species in the reduced state, and $Ag_m$ clusters. Thus, the second metal (promoter) plays a crucial role in the properties of the catalyst. Also the order of deposition of metals is relevant for the catalytic activity. Thus, Jouini H. et al. [31] reported on the reactivity of the catalyst CuFeZSM5 in $NH_3$-SCR of NO and analyzed the effect of the order of exchange of the metal. The characterization of CuFeZSM5 and FeCuZSM5 allowed the identification of metal species, which differ in quantity, environment and degree of aggregation depending on the metal deposition order, which leads to different catalytic behaviors in both catalysts.

Summarizing, the replacement of $Na^+$ cations with $Fe^{2+}$ and $Ag^+$ caused a decrease of the caustic modulus, which may indicate a modification of the Brønsted acidity. According to the elemental analysis, the selectivity of the exchange of various cations in the mordenite is as follows: $Ag^+ > Na^+ > Fe^{2+}$. According to XPS and UV–Vis spectra of bimetallic samples, silver exists in the form of reduced ($Ag_m$ and $Ag^0$ nanoparticles) and iron as $Fe^{3+}$ species, mainly, however these were not the cationic species that were used in solution for exchange with $Na^+$. Monovalent silver nitrate and bivalent perchlorate of iron were used as reagents for ion exchange. The appearance in the system of zero-valent silver and three-valent iron indicates the occurrence of a redox reaction by itself (Equation (1)), which was not expected. However, the electrochemical potentials of both species are very close making it possible that redox processes can be carried out ($Fe^{2+} \overset{e^-}{\Leftrightarrow} Fe^{3+} + 0.77$ and $Ag^+ \overset{e^-}{\Leftrightarrow} Ag^0 + 0.80$).

$$Ag^+ MOR \overset{Fe^{2+}, \, T, \, H_2O}{\rightarrow} \left( Ag^0 / Ag_m \right) Fe^{3+} MOR, \tag{1}$$

Finally, the bimetallic Ag-Fe/Mordenite catalysts showed dynamic behavior and different active sites during the reduction of $NO_x$ due to appearance of various metal species (according to spectroscopic results), such as clusters ($Ag_m^{n+}$, $Ag_m$ with m < 8), cationic species ($Ag^+$ and $Fe^{3+}$) and metal nanoparticles of Ag (with particle size less than 6 nm). We observed different species as isolated ion (monomeric), dimers, multimeric species and clusters in various oxidation states that may act as active centers in the zeolite framework and these materials can be considered as dynamic catalysts in the redox cycle for the NO reduction in presence of hydrocarbon under oxidizing conditions.

## 3. Materials and Methods

### 3.1. Sample Preparation

Mordenite type zeolite was supplied by Zeolyst International in sodium form (CBV-10A product, Zeolyst International, Conshohocken, PA, USA) with an atomic ratio Si/Al = 6.5 (Figure 6). The precursors of Ag and Fe were 0.1 N aqueous solutions of silver nitrate ($AgNO_3$, ≥99.0%, FagaLab, Sinaloa, México) and iron (II) perchlorate hydrate ($Fe(ClO_4)_2$ $H_2O$, ≥98.0%, Sigma-Aldrich, Saint Louis, MO, USA). Cations were introduced into mordenite by conventional ion exchange during 24 h at two different temperatures–ambient and 60 °C. After the ion exchange procedure, the samples were

filtered, washed with deionized $H_2O$ and dried at 110 °C for 20 h. All processing steps were carried out under conditions that prevent direct light from entering the samples, in order to avoid possible spontaneous reduction of silver ions. Monometallic systems were prepared by ion exchange of the original NaMOR in appropriate solution of Ag or Fe (Table 1). In the volume of solutions used to prepare mono- and bimetallic samples, the $Ag^+$ and $Fe^{2+}$ cations were in excess, that is, they contained more cations than it was necessary to complete exchange with the $Na^+$ cations. Therefore, no further changes depending on the volume of the solutions during the exchange for Ag or Fe cations in the zeolite are expected, unless the amount of the substance in the solution is reduced.

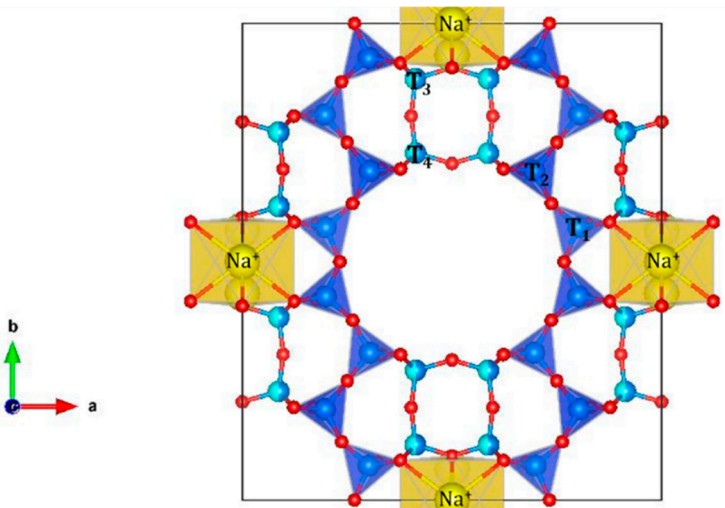

**Figure 6.** Mordenite viewed along [001] direction. The main channel is shown [12-ring 0.65 × 0.7 nm]. The $Na^+$ cations (yellow) and T-sites of NaMOR according to IZA database.

### 3.2. Sample Characterization

Elemental analysis of the samples was carried out by inductively-coupled plasma-optical emission spectroscopy (ICP-OES) in a Varian Vista-MPX equipment (Varian Inc., Palo Alto, CA, USA), using argon to generate plasma and nitrogen to clean the optical system.

The electronic transitions of the samples were studied by diffuse reflectance UV–Vis spectroscopy (DRS) using a Cary 100 spectrophotometer (Agilent Technologies Mexico, Ciudad de México, México) in the wavelength range 200–800 nm, with a resolution of 0.1 nm. The spectrum of the initial NaMOR was subtracted from the spectra of all samples.

The oxidation states of silver and iron on the surface were studied by XPS in a SPECS equipment with a Phoibos detector (DLD, HSA3500, SPECS, Berlin, Germany). A monochromatic Al-$K_\alpha$ X-ray source at 1486.6 eV was used for analysis. BE scale was calibrated using the Si 2p peak at 102.7 eV, the accuracy of the measurement was ±0.1 Ev. The spectra deconvolution was made by using the Casa XPS software (Version 2.3.19, Casa Software Ltd., Teignmouth, Devon, UK). Chemical states were determined on the basis of th areas and BE of Ag 3d and Fe 2p photoelectron peaks.

The morphology and particle size were determined from HR-TEM micrographs. The TEM measurements were carried out in a JEOL 2010 electron microscope (JEOL Ltd., Tokyo, Japan) operating at accelerating voltage 200 kV using $LaB_6$ filament. For analysis, the sample was dispersed in isopropanol using ultrasound, and putting a drop of this suspension on supported carbon film (lacey type). About 500 particles were chosen to determine the mean diameter of silver NPs. The mean particle diameter was calculated from constructed histograms.

*3.3. Catalytic Test*

The catalytic activity was evaluated in the $NO_x$ reduction in the presence of propene, carbon monoxide and oxygen atmosphere in a fixed bed continuous flow quartz micro-reactor inside a vertical furnace. Catalytic runs were performed using 100 mg of the catalyst packed in the U-shaped flow reactor, in the temperature range 25–500 °C with a ramp rate of 5 °C·min$^{-1}$. The reaction gas phase mixture consisted of NO [850 ppm], $C_3H_6$ [300 ppm], $O_2$ [2.1 vol%], CO [0.4 vol%], and $N_2$ as balance. The total flow was 7 mL·min$^{-1}$, corresponding to a mean hourly gas space velocity (GHSV) of 15,600 h$^{-1}$. Before the catalytic reaction, the catalysts were pretreated in a flow of the oxidant mixture (50 mL/min of 4.4 vol% of $O_2/N_2$ balanced) with a heating rate of 5 °C·min$^{-1}$ from room temperature up to 550 °C. Inlet and outlet gas concentrations were measured in an ZPA-IR gas analyzer (California Analytical Instruments Inc., Orange, CA, USA). Data acquisition of the NO concentration during the catalytic reaction was carried out continuously. The conversion of NO was calculated by means of the equation: NO conversion, $\% = [[NO_{in} - NO_{out}]/NO_{in}] \times 100$.

## 4. Conclusions

The characterization of the bimetallic catalysts allowed the identification of Fe and Ag ions and metal species with different degree of concentration, aggregation, and environment, which led to different catalytic behavior of the catalysts in the reduction of NO. The physicochemical analysis of bimetallic catalysts using UV–Vis, XPS, and HRTEM confirmed the presence of $Ag^0$ nanoparticles with an average particle diameter between 2 to 6.5 nm, the appearance of $Ag_m$ metal clusters and iron in an oxidation state III, mostly. These species varied with the order of exchange, i.e., Ag first then Fe or or Fe first, then Ag. Silver in bimetallic catalysts was mainly in a reduced state due to the influence of the iron cation transition $Fe^{2+}/Fe^{3+}$. In such a case, the mordenite framework was also decisive for the occurrence of that redox mechanism.

The AgFe, FeAg, and mAgFe bimetallic catalysts prepared at room temperature and 60 °C showed a different synergistic effect in their catalytic properties which was related with the preparation conditions. Each of three studied bimetallic catalysts showed different active sites during the NO reduction due to the presence of active species, such as $Fe^{3+}$ stabilized in the mordenite framework, $Ag^+$ cations and silver in reduced state ($Ag_m$ clusters and $Ag^0$ NPs). For all bimetallic catalysts, the NO conversion exceeded 70% in a temperature range between 300–450 °C.

As a result of the study, we concluded that mAgFeMOR catalysts in which both cations were incorporated simultaneously, gave the best catalytic results for NO reduction, at both temperatures, with 80% of conversion at 360 °C.

**Author Contributions:** P.S.-L. is the original draft writer, data curator, and researcher; Y.K. curated some data, researched, reviewed, and edited the manuscript; S.M. was responsible for conceptualization, article reviewing, and editing; F.C.-R. curated some data, acquired funding, and supervised, as well as reviewed and edited the paper; S.F. was primarily responsible for funding acquisition, and was also responsible for resources, investigation, supervision, reviewing, and editing of the manuscript; and V.P. was the main supervisor, and also responsible resources, methodology, investigation, reviewing, and editing of the manuscript.

**Funding:** This work was supported by RFBR and CITMA grant no. 18-53-34004 and UNAM-DGAPA-PAPIIT through grant IN107817. Perla Sánchez-López thanks to project SENER-CONACyT-Hidrocarburos for 117373 for scholarship support. Fernando Chávez-Rivas thanks the support of COFAA-IPN.

**Acknowledgments:** The authors thank I. Rodriguez-Iznaga for fruitful discussions and to A. Simakov, E. Smolentseva, E. Aparicio, D. Domínguez, M. Estrada, F. Ruiz, E. Flores, and A. Arteaga for technical assistance.

**Conflicts of Interest:** The authors declare no conflict of interest.

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
