# Peer review of "Bimetallic AgFe Systems on Mordenite: Effect of Cation Deposition Order in the NO Reduction with C3H6/CO"

_catalysts, doi:10.3390/catal9010058_

Round 1
Reviewer 1 Report
In the present manuscript, the authors describe the preparation and application of modified sodium mordenite zeolites as catalytsts for the gas phase reduction of NOX. The zeolites are properly modified with metals as Ag and Fe, through an ion exchange method. Mono- and di-metallic species are prepared at two different temperatures, modifying the order of deposition too.
In the introduction a good background on the state of the art and motivations are described. Page 1 raw 44 and following raw reports some acronyms (BEA, MOR, FER) that should be explicitated in the text.
Sanchez-Lopez et al. well describe the synthetic parameters and the subsequent characterization.
Regarding the synthesis of the materials, in my opinion Table 4 needs some revisions. The accessible limit of metal content (atomic %) is always constant for the two metals (3.2 for Fe and 6.4 for Ag) even though in the monometallic species 104 mL of metal precursor solution are used and in the bi-metallic species only 52 mL are employed. Is the accessible limit dependent on the amount of metal precursor? What do you mean by accessible limit? If this parameter is dependent only on the zeolite structure, please clarify it better in the text.
A good description of the materials is provided in the results section. In Table 1, I suggest the authors to explicit that A and 60 reported at the end of the materials' name is referred to the temperature of the synthesis. And I also suggest to add a first column in which you distringuish between room temperature and 60°C, so that the information is soon understood by the reader.
Starting from the experimental results the authors highlight that the loadings of both metals increase in the monometallic samples moving from room temperature to 60°C. In the bimetallic samples the same observation cannot be stated, as the authors clearly claim in the text. Did you expect this result? Why the loading is not improved by the increased temperature in the bi-metallic species? I would have expected the same result. I would suggest to add some more description of this experimental observation.
In the UV-Vis characterization, the authors claim that three silver species can be detected. The same is confirmed by XPS. I would try to add some quantitative data about the ratios between these species in the samples, if possible, especially starting from XPS data.
Page 4 raw 139, please explicit BE in the text as binding energy and put BE in the brakets, to improve the readiness of the text.
In the HRTEM images, the reduction of Ag particles is detected when Fe is also added to the modified zeolite, compared to the Ag monometallic sample. A reduction of Ag agglomeration is supposed to be responsible of the reduced Ag size. This observation makes sense when Fe is first added. In my opinion, it can be less intuitive when Ag is first added. How do the authors explain this experimental observation?
In Figure 3, I would change the colour of FeMORTa and FeMORT60, because it can be hardly seen in the printed version.
Considering all the previous considerations, the manuscript represents a good work in the field of catalysis, aiming at the development of highly performant catalytic materials. The manuscript fits with the aim of this journal and can be considered for publication after minor revisions.
Author Response
Comment 1
In the present manuscript, the authors describe the preparation and application of modified sodium mordenite zeolites as catalytsts for the gas phase reduction of NOX. The zeolites are properly modified with metals as Ag and Fe, through an ion exchange method. Mono- and di-metallic species are prepared at two different temperatures, modifying the order of deposition too. In the introduction a good background on the state of the art and motivations are described.
Page 1 row 44 and following row reports some acronyms (BEA, MOR, FER) that should be explicitated in the text.
We thank for this proposal of technical improvements; all the mentioned acronyms were explained. The next changes were made:
· “iron on beta zeolite (Fe/BEA)” instead of “Fe/BEA” – PAGE 1 ROW 44
· “iron on MFI zeolite (Fe/ZSM5)” instead of “Fe/ZSM5” – PAGE 2 ROW 45
· “iron on ferrierite zeolite (Fe/FER)” instead of “Fe/FER” – PAGE 2 ROW 45
· “iron on mordenite zeolite (Fe/MOR)” instead of “Fe/MOR” – PAGE 2 ROW 46
· “iron on faujasite zeolite (Fe/Y)” instead of “Fe/Y” – PAGE 2 ROW 46
Comment 2
Sanchez-Lopez et al. well describe the synthetic parameters and the subsequent characterization. Regarding the synthesis of the materials, in my opinion Table 4 needs some revisions. The accessible limit of metal content (atomic %) is always constant for the two metals (3.2 for Fe and 6.4 for Ag) even though in the monometallic species 104 mL of metal precursor solution are used and in the bi-metallic species only 52 mL are employed. Is the accessible limit dependent on the amount of metal precursor? What do you mean by accessible limit? If this parameter is dependent only on the zeolite structure, please clarify it better in the text.
The accessible limit of metal content is defined in zeolites as an exchange capacity, and is related to the amount of Na+ cations in the zeolite framework that are available for the exchange by other cations (in this case Ag or Fe). However, to clarify it for readers we exchanged “accessible limit” to “ion exchange capacity” in the text of the manuscript and in the table. Thus that exchange capacity is a constant value determined only by the Si/Al ratio that defines the total negative charge in the structure, and is independent on the nature of counter ions. Thus, this value can be derived from the elementary composition.
In the volume of solutions used to prepare mono- and bimetallic samples, the Ag and Fe cations were in excess, that is, they contained more cations than are necessary for a complete exchange with the Na cations. Therefore, no further changes depending on the volume of the solutions during the exchange for Ag or Fe cations in the zeolite are expected, unless the amount of the substance in the solution is reduced. This explanation was added into the text of the manuscript (PAGE 12 ROWS 325-329).
Thanks to Reviewer’s comment, it became clear the importunes to name the synthesis parameters under the study and to explain the sample’s labels before presenting the study results (PAGE 2 ROWS 65-73). Table was modified and moved to the beginning of the manuscript (See Table 1).
Comment 3
A good description of the materials is provided in the results section. In Table 1, I suggest the authors to explicit that A and 60 reported at the end of the materials' name is referred to the temperature of the synthesis. And I also suggest to add a first column in which you distinguish between room temperature and 60°C, so that the information is soon understood by the reader.
We thank for this suggestion of technical improvements; all the mentioned corrections were made. Table 2 (before Table 1) was modified, including the new column of temperature used during ion exchange and, as it was already mentioned, the sample’s label explanation was placed before results.
Comment 4
Starting from the experimental results the authors highlight that the loadings of both metals increase in the monometallic samples moving from room temperature to 60°C. In the bimetallic samples the same observation cannot be stated, as the authors clearly claim in the text. Did you expect this result? Why the loading is not improved by the increased temperature in the bi-metallic species? I would have expected the same result. I would suggest to add some more description of this experimental observation.
Although the study of ion exchange kinetics was not the goal of this work. In accordance with our previous results [Microporous Mesoporous Mater., V. 255, pp. 200–210, 2018 – This Ref. was added to the manuscript as a # 50] increasing the temperature of ion exchange, we expected an increase in the content of Ag and Fe cations due to the acceleration of diffusion, which was actually observed for monometallic samples. However, in a bimetallic catalyst, competition for the exchange sites plays a critical role cause of the cation sizes are close to the diameter of the zeolite channel. However, this process certainly needs additional research, the results of which will be published elsewhere. The corresponding explanation was added to the text (PAGE 3 ROWS 121-128)
Comment 5
In the UV-Vis characterization, the authors claim that three silver species can be detected. The same is confirmed by XPS. I would try to add some quantitative data about the ratios between these species in the samples, if possible, especially starting from XPS data.
Table 3 (before Table 2) was modified, it shows the contribution of each species Ag+ and Ag0 on the surface, calculated from the deconvolution of the peaks.
Comment 6
Page 4 ROW 139, please explicit BE in the text as binding energy and put BE in the brackets, to improve the readiness of the text.
Correction applied (PAGE 5 ROW 168).
Comment 7
In the HRTEM images, the reduction of Ag particles is detected when Fe is also added to the modified zeolite, compared to the Ag monometallic sample. A reduction of Ag agglomeration is supposed to be responsible of the reduced Ag size. This observation makes sense when Fe is first added. In my opinion, it can be less intuitive when Ag is first added. How do the authors explain this experimental observation?
In both cases (Ag+Fe and Fe+Ag) the reduction of Ag particle size was independent on the of cation deposition order; this is due to the fact that iron (II) species interact with silver (I) cations, causing the formation of smaller silver nanoparticles (Ag NPs). This interaction is represented by Equation 1.
Comment 8
In Figure 3, I would change the color of FeMORTa and FeMORT60, because it can be hardly seen in the printed version.
We thank for this proposal of technical improvements, Figure 1 and 5(before Figure 2 and Figure 3) were corrected in accordance with the recommendation of the Reviewer.
Reviewer 2 Report
The manuscript entitled Bimetallic AgFe systems on mordenite: effect of cation deposition order on the NO reduction with C3H6/CO describes experimental results obtained with catalysts synthesised with Ag and Fe as mono and bimetallic active phases. Divers techniques have been used to characterise their differences and relate them to the results obtained of NO reduction in a fixed bed reactor. The subject is suitable for publication in Catalysts and the paper presents novel research work in a clear and well organised way. The English of the paper is very satisfactory. The abstract is concise and informative. There is a high number of references, some of them very recent. Some major points could be improved before publication:
Why Fe particles do not appear in the HRTEM micrographs? Could you introduce an explanation in the text if missed?
The authors could introduce XRD to know the crystallite size of Ag and Fe in the catalysts what is always a relevant characteristic related to activity. The authors could also study as well the N2 adsorption of the samples. The variations in the N2 adsorption produced in the synthesis process are also an important factor related to zeolite activity.
More results related to the activity of the catalyst could be described in the manuscript like C3H6 conversion, CO conversion and NO conversion into N2 with temperature. Their study could bring light into the explanation of the different behaviour produced by the catalysts that could be introduced in the text
What do they mean the arrows placed in Figure 3?
How did you choose the synthesis temperatures ambient and 60 ºC?
The conclusions section does not clearly offer the effect of the order of the metal deposition on the catalytic activity. Which order produces higher NO conversion into N2 at lower temperatures, Ag+Fe or Fe+Ag ?
Author Response
Comment 1
Why Fe particles do not appear in the HRTEM micrographs? Could you introduce an explanation in the text if missed?
As it is mentioned in experimental part, no reduction treatments were undertaken; cationic silver is reduced by oxidizing Fe(II) to Fe(III). The HRTEM results confirm that Fe cations are well dispersed in the zeolite framework without agglomeration.
Comment 2
The authors could introduce XRD to know the crystallite size of Ag and Fe in the catalysts what is always a relevant characteristic related to activity. The authors could also study as well the N2 adsorption of the samples. The variations in the N2 adsorption produced in the synthesis process are also an important factor related to zeolite activity. More results related to the activity of the catalyst could be described in the manuscript like C3H6 conversion, CO conversion and NO conversion into N2 with temperature. Their study could bring light into the explanation of the different behavior produced by the catalysts that could be introduced in the text.
The average diameters of silver particles are shown in Table 4. We do not expect formation of Fe metal particles. We sincerely thank the Reviewer for an interesting proposal for the development of additional research with the system understudy.
Comment 3
What do they mean the arrows placed in Figure 3?
The arrows denote areas associated with different active species; however, Figure 5 (before Figure 3) was modified, and these arrows were eliminated.
Comment 4
How did you choose the synthesis temperatures ambient and 60 ºC?
The temperature increase is proposed as a parameter promoting the diffusion of ions into the zeolite in accordance with recommended practices (Ref. # 50 of the manuscript).
Comment 5
The conclusions section does not clearly offer the effect of the order of the metal deposition on the catalytic activity. Which order produces higher NO conversion into N2 at lower temperatures, Ag+Fe or Fe+Ag ?
As a result of the study, we came to the conclusion that mAgFeMOR catalysts, in which both cations were incorporated simultaneously, at both temperatures, gave the best catalytic results for NO reduction with 80% of conversion at 360 °C. By comparing the catalytic results of the samples with different deposition order, we conclude that AgFeMOR catalysts were more actives than the FeAgMOR ones, presenting high conversion at lower temperatures.
Therefore, bimetallic catalysts showed different active sites during the NO reduction due to the presence of active species such as Fe3+ stabilized in the mordenite framework, Ag+ cations and silver in reduced state (Agm clusters and Ag0 NPs).
The second conclusion about the influence of the order of the component deposition on the catalytic properties was modified and split in two (see PAGE 13 ROWS 374-382).

Reviewer 3 Report
This manuscript describes the preparation of bimetallic catalyst
FeAgMOR, by ion exchange of Na+ cation in mordernite, and their
application as catalyst in NOx reduction. The experimental work is well
and appropriately performed and the results very interesting. The
manuscript is clear and easy to follow, but in my opinion there are some
pending issues than the authors must make clear before being ready to be
published in “Catalysts”
With the data included in the manuscript I cannot see the direct
relation between the presence of iron (III) and the reduction of Ag(I)
to Ag(0). Both processes are expected and easily understandable under
the experimental conditions chosen for the preparation of the bymetallic
materials.
The UV-vis spectra for FeMORTa and FeMORT60 show an additional band
above 325 nm which has not been assigned. This is important due to the
assignment of the wide band around 325 nm in AgFeMOR, FeAgMOR and
mAgFeMOR bimetallic systems to silver metallic nanoparticles and the
subsequent discussion. On the other hand, the decrease of the band due
to Ag+ in this spectra is expected, since as the authors have explained,
the Ag cations are easily exchanged into the mordenite matrix. This
point must be further discussed and supported with experimental data,
anyway, it is just a proposal…
XPS figures should be included in the manuscript in order to make clear
the discussion of the data.
The 3d5/2 peak position of the main component (80%) attributed to Ag0
shifts also 0.3 eV in Ag60. So the formation of small silver
nanoparticles in bimetallic materials cannot be supported exclusively by
this data. I agree with this final statement: “_For all samples_ the BE
from the Ag spectra showed that ionic silver is reduced and is present
mainly on the surface as Ag-NPs and iron appears like oxides in
intermediate states in both cases”
On the basis of HRTEM-Micrographs I can see the influence of the
synthesis by steps, that is the exchange reaction with one metal after
another seems to affect the amount of loading metal and metal particle
distribution on the support; but the one step exchange with a mixture of
metals seems to guarantee the smallest silver particle distribution and
to avoid agglomeration, as well.
In my opinion the material mFeAgMOR has not been properly discussed in
comparison with AgFeMOR, FeAgMOR. It seems that gives high conversion
values at lower temperatures and in addition offers the advantage of
being easily synthesized in one step procedure.
Some minor comments
After the ion exchange with Ag and Fe this ratio decreased for both
cases Fe2+ and Fe3+ (See Table 1). Which entry do you mean in table 1?
The authors must mentioned the different entries in the tables..
Why Fe(ClO_4 )_2 was used instead of Fe(NO_3 )_2 ?
Molarity is a much more appropriate concentration unit than normality
Author Response
We thank you for confirming that our manuscript is of interest to a wide range of experts in this field, and we hope that after publication our work will be demanded by the scientific community.
Comment 1
With the data included in the manuscript I cannot see the direct relation between the presence of iron (III) and the reduction of Ag(I) to Ag(0). Both processes are expected and easily understandable under the experimental conditions chosen for the preparation of the bimetallic materials.
Monovalent silver nitrate and bivalent iron perchlorate were used as reagents for ion exchange. By itself, the appearance in the system of zero-valent silver and three-valent iron indicates the occurrence of a redox reaction. The corresponding explanation was added into the text of the manuscript PAGE 11 ROWS 300-305.
Comment 2
The UV-Vis spectra for FeMORTa and FeMORT60 show an additional band above 325 nm which has not been assigned. This is important due to the assignment of the wide band around 325 nm in AgFeMOR, FeAgMOR and mAgFeMOR bimetallic systems to silver metallic nanoparticles and the subsequent discussion. On the other hand, the decrease of the band due to Ag+ in this spectra is expected, since as the authors have explained, the Ag cations are easily exchanged into the mordenite matrix. This point must be further discussed and supported with experimental data, anyway, it is just a proposal…
The band above 325 nm related to silver metallic nanoparticles was not observed for FeMORTa and FeMORT60 (for the case it is not clear from the Figure, we mention that on the PAGE 4 ROWS 154-155).
Comment 3
XPS figures should be included in the manuscript in order to make clear the discussion of the data.
We thank for this suggestion of technical improvements; the mentioned figure was added as Fig.2 and 3.
Comment 4
The 3d5/2 peak position of the main component (80%) attributed to Ag0 shifts also 0.3 eV in Ag60. So the formation of small silver nanoparticles in bimetallic materials cannot be supported exclusively by this data. I agree with this final statement: “_For all samples_ the BE from the Ag spectra showed that ionic silver is reduced and is present mainly on the surface as Ag-NPs and iron appears like oxides in intermediate states in both cases”
The formation of small silver nanoparticles in bimetallic materials is supported not by XPS data exclusively, but also by UV-Vis data (see PAGE 4 ROWS 139-143). Thanks to this comment, we found it important to modify the UV-Vis data description (PAGE 4 ROWS 146-149, 158-162) and Table 3 to clarify presented data for a reader.
Comment 5
On the basis of HRTEM-Micrographs I can see the influence of the synthesis by steps, that is the exchange reaction with one metal after another seems to affect the amount of loading metal and metal particle distribution on the support; but the one step exchange with a mixture of metals seems to guarantee the smallest silver particle distribution and to avoid agglomeration, as well.
The Reviewer’s observation was based on HRTEM-Micrographs from Figure 4 (before Figure 2), which includes just 3 Ag-content samples of 8 ones considered in the manuscript. The complete results of the silver nanoparticle average diameter are presented in Table 4 (before Table 3), which were considered below. Thanks to this comment, we found it necessary to move Table 4 and its description next to the Fig. 4 and add some explanation (PAGE 8 ROWS 197 - 203).
Comment 6
In my opinion the material mFeAgMOR has not been properly discussed in comparison with AgFeMOR, FeAgMOR. It seems that gives high conversion values at lower temperatures and in addition offers the advantage of being easily synthesized in one step procedure.
We agree with Reviewer’s comment; the conclusions about the influence of the order of the component deposition on the catalytic properties was modified (PAGE 13 ROWS 374-382).
Comment 7
After the ion exchange with Ag and Fe this ratio decreased for both cases Fe2+ and Fe3+ (See Table 1). Which entry do you mean in Table 1? The authors must have mentioned the different entries in the tables…
We thank for this suggestion of technical improvements; the mentioned correction was made to the Table 1 description (PAGE 3 ROWS 96 and 109).
Comment 8
Why Fe(ClO_4 )_2 was used instead of Fe(NO_3 )_2 ?
Iron (II) nitrate as several other salts of Fe(II) are exposed to hydrolysis processes, their application required a pH control by the addition of corresponding acid. It makes more complex both the synthesis and the study of those materials. One of the stable to hydrolysis precursors of Fe(II) is Fe(ClO4)2 was used in the present work.
Comment 9
Molarity is a much more appropriate concentration unit than normality
Molarity (M) is the concentration of a solution expressed as the number of moles of solute per liter of solution. Normality is a measure of concentration equal to the gram equivalent weight per liter of solution. Gram equivalent weight is the measure of the reactive capacity of a molecule. The solute's role in the reaction determines the solution's normality. Both of these units are equal when measuring the concentration of solutions. Normality is more appropriate in the case of ion-exchange synthesis especially for the different charge cations cause of the constant charge of the zeolite ion-exchange centers.

Round 2
Reviewer 2 Report
Thorough responses from the authors to the reviewer comments have been made. Therefore, the modified paper can be published